# Surface-Confined Self-Assembly of Star-Shaped Tetratopic Molecules with Vicinal Interaction Centers

**DOI:** 10.3390/molecules30122656

**Published:** 2025-06-19

**Authors:** Jakub Lisiecki, Damian Nieckarz

**Affiliations:** 1Jerzy Haber Institute of Catalysis and Surface Chemistry, Polish Academy of Sciences, Niezapominajek 8 Street, 30-239 Kraków, Poland; jakub.lisiecki@ikifp.edu.pl; 2Department of Theoretical Chemistry, Institute of Chemical Sciences, Faculty of Chemistry, Maria Curie-Skłodowska University in Lublin, Maria Curie-Skłodowska Square 3, 20-031 Lublin, Poland

**Keywords:** Monte Carlo simulations, self-assembly, *vicinal* functional groups, adsorbed overlayers

## Abstract

Precise control over the morphology of surface-supported supramolecular patterns is a significant challenge, requiring the careful selection of suitable molecular building blocks and the fine-tuning of experimental conditions. In this contribution, we demonstrate the utility of lattice Monte Carlo computer simulations for predicting the topology of adsorbed overlayers formed by star-shaped tetratopic molecules with *vicinal* interaction centers. The investigated tectons were found to self-assemble into a range of structurally diverse architectures, including two-dimensional crystals, aperiodic mosaics, Sierpiński-like aggregates, and one-dimensional strands. The theoretical insights presented herein deepen our understanding of molecular self-assembly and may aid in the rational design of novel nanomaterials with tunable porosity, chirality, connectivity, and molecular packing.

## 1. Introduction

Surface-confined self-assembly is a process in which molecules organize into complex, well-defined patterns that exhibit emergent properties beyond those of the individual components [1,2]. This phenomenon plays a central role in surface supramolecular chemistry, enabling the bottom-up fabrication of low-dimensional architectures that rival the structural sophistication of naturally occurring systems [3,4,5,6]. Unlike the top-down approach, the self-assembly outcome of adsorbed molecules is highly sensitive to their intrinsic properties—such as size, shape, and functionalization scheme—as well as external parameters, including temperature T, surface coverage θ, and the type and symmetry of the substrate [7,8,9]. When these factors are carefully tuned, it becomes possible to direct the formation of atomically thin supramolecular constructs with tailored porosity, composition, molecular packing, and connectivity [10,11,12,13].

However, the rational design of such assemblies is challenging, as the synthesis and purification of new molecular tectons is both time-consuming and resource-intensive. A promising strategy to overcome this limitation involves the use of fast, predictive computer simulations, in which functional molecules are modeled in a simplified yet chemically meaningful way [14,15,16,17,18]. While this approach has been widely applied to tripod-like molecules bearing spatially isolated functional groups (FGs) [19,20,21,22], analogous studies focusing on star-shaped tectons with *vicinal* FGs have not been conducted to date.

Therefore, in this work, we use lattice Monte Carlo (MC) simulations to predict the self-assembly of star-shaped positional isomers (regioisomers), each of which has two *vicinal* interaction centers, and bears a total of four identical FGs (Figure 1). Compared to other popular computational techniques such as density functional theory (DFT) [23] and molecular dynamics (MD) [24], the coarse-grained MC simulations are historically the oldest, conceptually the simplest, and unrivaled in terms of computational efficiency [25].

In our coarse-grained model, the flat triangular lattice serves as an idealized representation of a face-centered cubic (fcc) metallic surface and plays a dual role: it confines molecular assembly to two dimensions and facilitates the formation of supramolecular architectures stabilized by anisotropic yet reversible intermolecular interactions. In this context, the vertices of the triangular lattice represent energetically equivalent adsorption sites, each of which can either remain vacant or be occupied by a single molecular segment.

## 2. Results and Discussion

Let us begin our discussion with molecule **A**, which features the simplest functionalization scheme among all tectons investigated in this study. As shown in the top-left inset of Figure 2, the *vicinal* FGs of molecule **A** are positioned on its top arm, oriented upward and to the right. In contrast, the remaining arms of **A** are functionalized exclusively at the *para* positions. Consequently, all FGs of tecton **A** are directed outward from its conformationally rigid, tripodal backbone, suggesting that it may be well-suited for surface-assisted self-assembly. Surprisingly, preliminary simulation results obtained for molecule **A** appear to contradict this tentative thesis. Specifically, at moderate surface coverage (0.33≤θ≤0.36), we observed the selective formation of an aperiodic phase, denoted A^H^, with an average density $\langle \rho^{H}\rangle \approx 0.41$ (Figure 2).

Statistical analysis of ten independent system replicas, each containing 1000 molecules **A**, revealed that more than one-fifth (21.78% ± 0.44) of tectons **A** remain coordinatively unsaturated at temperature T=0.01. This intriguing observation prompted us to perform additional simulations for molecule **A** at lower surface coverages (θ<1/3). Surprisingly, reducing the surface coverage dramatically altered the self-assembly outcome of molecules **A**. For instance, at θ=0.26, we observed the emergence of two highly ordered polymorphs, denoted A^I^ and A^II^ (Figure 3).

As shown in Figure 3a, the defect-free network A^I^ ($\rho^{I}=0.39$) consists of two types of nanopores: achiral hexagonal voids α (19 free adsorption sites, diameter d=6, measured corner-to-corner) and R-chiral cavities β (each comprising 6 adsorption sites). This biporous network possesses a centrosymmetric rhombic unit cell ($\sqrt{61}\times \sqrt{61},\;∠\;60/120^{o}$). In this double-walled phase, all tetratopic molecules are interconnected exclusively via the ***p****p*-***p****p*-***v****vv* mode: each *para* (***p***) FG of a tecton **A** is bonded to a *para* (*p*) FG of another molecule, while its *vicinal* (***v***) FGs interact with *vicinal* (*v*) FGs of two neighboring molecules. This concise shorthand notation will be used throughout the text.

In contrast to the biporous phase A^I^, the ordered phase A^II^ ($\rho^{II}=0.46$) consists of molecules **A** connected exclusively in a ***p****v*-***p****v*-***v****pp* configuration and features a smaller rhombic unit cell ($2\sqrt{13}\times 2\sqrt{13},\;∠\;60/120^{o}$), see Figure 3b and Figure 4. Notably, at surface coverages θ≤0.20, the hybrid phase A^H^ was not observed. Instead, the periodic networks A^X^ (where X is I or II) were formed selectively. Polymorph A^II^ appeared in 43 out of 60 simulated systems (~72%). The origin of this bias lies in the potential energy difference between competing phases A^X^ at temperatures T≥0.4 (Figure 5, left panel). At a moderate surface coverages (0.20<θ≤0.30), the self-assembly outcome of molecules **A** became less predictable: depending on the simulation, we observed the exclusive formation of a single phase A^X^, coexistence of polymorphs A^I^ and A^II^, or the emergence of the hybrid phase A^H^. Interestingly, within the surface coverage range $\langle \rho^{H}\rangle <\theta \leq \rho^{II}$, the hybrid phase A^H^ was not formed (Figure 4).

The relationship between the fraction f (0≤f≤1) of under-coordinated molecules **A** and surface coverage θ is illustrated in the right panel of Figure 5.

At low surface coverages (θ<1/5), the fraction f decreases monotonically, following a power-law dependence on θ:$$f\approx \frac{5}{L\sqrt{\theta }}$$
where L is the linear size of the simulation box. As θ increases further, f begins to rise, and eventually reaches a relatively broad plateau. The selective formation of the hybrid phase A^H^ within this surface coverage range can be tentatively explained as follows: the absolute average total potential energy $\left|\langle E^{H}\rangle \right|$ of the hybrid overlayer A^H^ is approximately 3% lower than that of the isoenergetic networks A^I^ and A^II^ (Table 1). However, the high entropy of the aperiodic phase A^H^ favors its formation in the range $0.30\leq \theta \leq \langle \rho^{H}\rangle$. For $\theta >\langle \rho^{H}\rangle$, there is insufficient space for the lateral growth of the structurally heterogeneous phase A^H^, and the relatively close-packed network A^II^ forms almost exclusively (Figure 4).

Like molecule **A**, the next investigated tectons, **B** and **C**, also proved to be network-forming. In particular, the supramolecular architecture formed by molecules **B** (Figure 6a) is more intricate than the highly ordered networks generated by molecules **A** (Figure 3). The discussed overlayer is composed of patches of ordered network $B^{\gamma }$ (ρ=4/7, $\sqrt{7}\times \sqrt{7},\;∠\;60/120^{o}$, **p**v**-m^R^**v**-v**m^R^p), interconnected by the rows of R-chiral voids β and achiral cavities δ.

As illustrated in Figure 6b, the two-dimensional (2D) carpet-like structure is composed of tectons **C** interconnected in the **m^S^**p**-p**m^R^**-v**vv mode. The resulting network exhibits moderate density (ρ=4/9), has a rhombic unit cell ($3\sqrt{3}\times 3\sqrt{3},\;∠\;60/120^{o}$), and contains two types of cavities that differ in size and shape: S-chiral pores γ and significantly larger pores $\gamma^{2}$ (12 adsorption sites each).

In contrast to the network-forming molecules **A**–**C**, tectons **D** was found to generate surprisingly complex supramolecular patterns characterized by a random distribution of triangular pores $\gamma^{n}$ of various sizes. Here, the superscript n denotes the number of molecules **D** forming each side of the triangular pore $\gamma^{n}$ (Figure 7).

Among the triangular pores $\gamma^{n}$, the R-chiral voids γ are the most numerous and often form the walls of larger cavities: $\gamma^{2}$ (12 adsorption sites), $\gamma^{3}$ (30 adsorption sites), $\gamma^{4}$ (54 adsorption sites), $\gamma^{5}$ (85 adsorption sites), and others. Interestingly, all empty pores $\gamma^{n}$ embedded within the mosaic-like pattern shown in Figure 7a are oriented in the same direction—a consequence of the anisotropic lateral growth of the aperiodic overlayer. Moreover, the walls of most pores $\gamma^{n}$ are composed of coordinatively saturated molecules **D**. As a result, the percentage of under-coordinated molecules in this adsorbed system is relatively low (Table 2).

Interestingly, the overlayer depicted in Figure 7a bears a striking resemblance to the staircase mosaic pattern found on the shell of the venomous sea snail *Conus textile* [26]. This similarity mainly arises from the presence of undirected, variably sized voids $\gamma^{n}$ that form a reticulated pattern composed of repeating geometrical motifs. In this context, molecule **D** can be regarded as a molecular analogue of Stephen Wolfram’s cellular automata governed by rules 30 and 110 [27]. Moreover, the aperiodic mosaic depicted in Figure 7b is complemented by closed supramolecular aggregates resembling successive iterations of the deterministic fractal known as the Sierpiński triangle (ST) [28]. Notably, these Sierpiński-like triangles are not only visually striking but also structurally scalable. For instance, the nth ST iteration (where n=1,2,3, …) consists of $\left({27\cdot 3}^{n-1}+3\right)/2$ molecules **D**, contains ${10\cdot 3}^{n-1}$ empty pores γ, and includes $\left({7\cdot 3}^{n}-1\right)$/2 triangular voids in total.

Compared to the fractal-forming molecule **D**, the functionalization scheme of the next studied tecton (**E**) is even more complex. Specifically, three of its four FGs are parallel to each other, facilitating the formation of two distinct networks, E^I^ and E^II^ (Figure 8). The brickwall-type polymorph E^I^ (ρ=1/2, **p**p**-m^S^**v**-v**m^S^v) has a relatively simple structure, as it comprises parallel, alternating strips of S-chiral pores β and achiral voids δ. Its parallelogram unit cell ($5\times \sqrt{7},\;∠\;80/100^{o}$) encloses only two molecules **E** and is significantly smaller than the rhombic unit cell of the flower-like network E^II^ ($2\sqrt{13}\times 2\sqrt{13},\;∠\;60/120^{o}$, ρ=8/17, **p**v-**m**^S^v-**v**pm^S^) shown in Figure 8b. Moreover, the likelihood of occurrence of the competing polymorphs E^I^ and E^II^ is not equal, as the floral network E^II^ was selectively obtained in only 15 out of 50 system replicas.

For molecule **F**, our simulations reveal the formation of a single nanoporous network, shown in Figure 9a. This crystalline phase has a rhombic unit cell ($\sqrt{43}\times \sqrt{43},\;∠\;60/120^{o}$) and a moderate density (ρ=4/7, **m^R^**v**-m^R^**v**-v**m^R^m^R^), and consists of R-chiral pores γ along with cogwheel-like voids η (13 adsorption sites, diameter d=4, each).

In the next studied system, we observed the formation of parallel supramolecular strands composed of tectons **G**, connected via the **m^S^**v**-m^S^**v**-v**m^S^m^S^ mode (Figure 9b). These strands vary in length and contain periodically distributed pores δ. Their parallel arrangement facilitates unhindered elongation, thereby minimizing the system’s total potential energy. Interestingly, at twice the surface coverage (θ≈2/3), the porous strands exhibit a tendency to align side-to-side, despite the absence of attractive lateral interactions between them (Figure 10). This alignment results in a 2D uniporous phase with an extremely high density (ρ=8/9) and a small parallelogram unit cell ($3\times \sqrt{7}$).

Figure 11 presents the simulation results for molecules **H**, which can form two competing porous networks. A distinctive feature of this anchor-like building block is that its arms bear two *meta* FGs arranged at a $120^{o}$ angle. As a result, the lamellar overlayer H^I^ (ρ=8/13, **m^R^**m^R^**-m^S^**v**-v**m^S^v) (Figure 11a) resembles the biporous phase E^I^ shown in Figure 8a. Specifically, the ordered pattern H^I^ consists of parallel rows of δ and κ pores (four adsorption sites each) and has a parallelogram unit cell ($\sqrt{3}\times \sqrt{21},\;∠\;76/104^{o}$). Interestingly, the second polymorphic phase H^II^ (bonding mode: **m^R^**v**-m^S^**v**-v**m^R^m^S^) (Figure 11b) is structurally more complex than H^I^ and is isomorphous with the floral network illustrated in Figure 8b.

For the last studied molecule (**I**), we also observed the formation of two polymorphic networks that differ markedly in porosity, density, and molecular connectivity. As shown in Figure 12a, the single-walled network I^I^ exhibits extremely low density ($\rho^{I}\approx 1/3$) and features large cogwheel-like voids λ (49 free adsorption sites, diameter $d=4\sqrt{3}$) whose rims are decorated with a garland of small pores δ. In the zoomed-in fragment of this openwork phase, the centrosymmetric rhombic unit cell ($\sqrt{78}\times \sqrt{78},\;∠\;60/120^{o}$) is marked and encloses six molecules **I** interconnected in the **m^R^**m^S^**-m^S^**m^R^**-v**vv mode.

Contrary to the reticular phase I^I^ (Figure 12a), the uniporous polymorph I^II^ ($\rho^{II}=2/3$, **m^R^**v**-m^S^**m^S^**-v**m^R^v) has a parallelogram unit cell ($4\times \sqrt{7},\;∠\;78/102^{o}$) and consists of long strands of antiparallel molecules **I** interconnected via anti-clockwise-rotated *meta* FGs. As shown in Figure 12b, ordered polymorph I^II^ can be selectively formed over a relatively broad surface coverage range (1/3<θ≤2/3), as $\rho^{II}\approx 2\rho^{I}$.

To gain deeper insight into structure formation in the investigated adsorbed systems, we calculated the temperature dependence of the fractions of molecules **A**–**I** connected to 0–4 nearest neighbors, as illustrated in Figure 13.

At high temperature (T=1), the fraction of free molecules (n=0) is approximately two-thirds for all positional isomers (**A**–**I**) bearing *vicinal* FGs. This fraction then decreases monotonically as the temperature is lowered, approaching zero near T=0.2. The curves for molecules connected with only one neighbor (n=1) similarly rise gradually to a maximum before rapidly declining to zero at T≈0.2 (with the exception of the strand-forming regioisomer **G**). The bell-shaped curves for n=2 feature a single, sharp, and roughly symmetric peak, although their widths vary noticeably. As shown in Figure 13, the unimodal curves for n=3 differ in shape: some (**A**, **B**, **D**, **F**, **G**) are Gaussian-like, while others (**C**, **E**, **H**, **I**) are significantly flattened and centered around T≈0.3. Fully coordinated molecules (n=4) are rare at higher temperatures (T≥0.4), but their fraction rises rapidly near T≈0.3, coinciding with large fluctuations in potential energy, as clearly shown in Figure 14. The specific heat c was defined as the heat capacity C per molecular segment and calculated using the equation:$$c=\frac{C}{4N}=\frac{\langle E^{2}\rangle -{\langle E\rangle }^{2}}{4Nk_{B}T^{2}}$$
where N is the total number of molecules, and the factor 4N means the total number of hard segments (each studied molecule consists of four hard segments), $k_{B}$ is the Boltzmann constant (set to unity in our model), and T is the system temperature.

As shown in Figure 14, all specific heat curves are relatively smooth at temperatures T≥0.5, indicating low potential energy fluctuations in this range. Moreover, each curve rises monotonically as the temperature decreases, reaches a more or less sharp peak corresponding to the lattice gas-ordered phase transition, and then rapidly decreases. Finally, at T=0.1, potential energy fluctuations almost completely vanish. This is not surprising, since the studied molecules (**A**–**I**) are positional isomers, with each possessing four sterically accessible FGs.

## 3. The Model and Calculations

The lattice MC computer simulations were conducted on a rhombic fragment of a triangular lattice mimicking an achiral, flat metallic (111) surface (Figure 15).

The distance between adjacent vertices (adsorption sites) of the triangular lattice was set to l=1. In our model, the arms of star-shaped tectons **A**–**I** were equipped with well-defined interaction centers representing FGs of real organic molecules. Adsorbed molecules were allowed to interact laterally with a potential energy ε=−1 whenever their outermost segments (arms) occupied neighboring adsorption sites and their assigned interaction directions were collinear and oppositely oriented (→←). All other intermolecular and molecule–surface interactions were neglected for simplicity (set to zero). Additionally, during the simulations, the flipping of the conformationally rigid tectons **A**–**I** on the triangular lattice was not allowed. All MC simulations were carried out in the canonical ensemble: the total number of admolecules N and the linear size of the rhombic simulation box (L=110) were held constant. To eliminate edge effects, periodic boundary conditions were imposed on the triangular lattice. The simulations began with a random distribution of N molecules on the triangular lattice at a temperature T=1.01. The modeled system was then linearly cooled to T=0.01, with the temperature range divided into 1000 equal intervals ∆T=0.001. To ensure energetic equilibration of the studied overlayers, a total of $5N\cdot 10^{8}$ MC steps were performed, each consisting of an attempt to displace and/or rotate a randomly selected molecule in-plane. For each attempt to displace a randomly chosen molecule, its potential energy $E_{o}$ was computed. The molecule was then removed from the simulation box and reassigned new coordinates (i, j), where 1≤i, j≤L, resulting in a trial configuration $\Omega_{n}$. The acceptance of $\Omega_{n}$ was determined by the standard Metropolis criterion [29]:$$p\left(\Omega_{o}\to \Omega_{n}\right)=min\left[1,exp\left(-\frac{∆E}{k_{B}T}\right)\right]$$
where $∆E=E_{n}-E_{o}$ is the energy difference between the new ($\Omega_{n}$) and old ($\Omega_{0}$) configuration, $k_{B}$ is the Boltzmann constant (set to unity), and T is the system temperature (0.01≤T≤1.01). If ∆E≤0, the new configuration $\Omega_{n}$ was accepted unconditionally. Otherwise, a random number $r\in \left[0,1\right]$ was generated and compared with the Metropolis probability p. If r≤p, $\Omega_{n}$ was accepted; otherwise, the molecule reverted to its previous position $\Omega_{o}$ on the triangular lattice. To assess the 2D packing of the ordered supramolecular constructs, we employed a dimensionless parameter called phase density, defined as$$\rho =\frac{2\sqrt{3}m}{\Lambda },\;\;\;\;\;0\leq \rho \leq 1$$
where m is the number of molecules in a unit cell and Λ is the total area of that unit cell. Similarly, surface coverage θ was defined as the ratio of occupied sites to the total number of adsorption sites in the rhombic simulation box ($L^{2}=12,100$). All simulations were conducted in a Linux environment on computational clusters at the Department of Theoretical Chemistry, Maria Curie-Skłodowska University in Lublin. On average, each simulation run lasted approximately 16 h and 20 min. Custom simulation codes were developed in the Fortran programming language.

## 4. Conclusions

In this work, we identified the key factors governing the surface-assisted self-assembly of star-shaped tectons (**A**–**I**) and provided a detailed description of the resulting overlayer topologies. All investigated molecules were found to be capable of forming coordinatively saturated supramolecular constructs differing in terms of porosity, connectivity, molecular packing, and density. Remarkably, we observed the formation of 2D homoporous and heteroporous networks, linear molecular strands, Sierpiński-like fractal aggregates, and aperiodic mosaics resembling natural patterns found on *Conus textile* sea snail shells. Overall, our findings can aid in the preliminary selection of star-shaped building blocks with *vicinal* functional groups for the targeted fabrication of supramolecular architectures with tailored structural properties. Furthermore, the theoretical approach proposed herein can be readily extended to racemic mixtures of prochiral tectons (**A**–**I**), as well as to their analogues with reduced backbone symmetry.

## Figures and Tables

**Figure 1 molecules-30-02656-f001:**
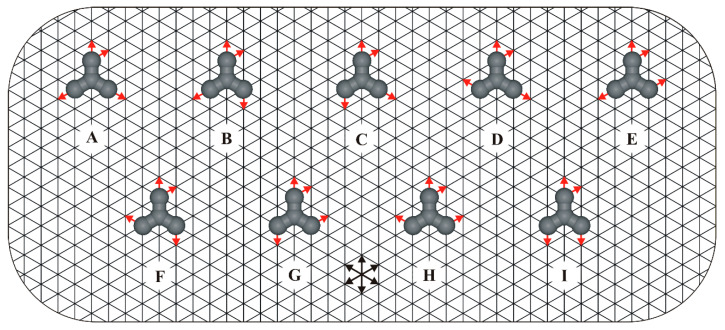
Family of star-shaped regioisomers (**A**–**I**) adsorbed on a triangular lattice. For clarity, only one surface enantiomer (R) is shown for each prochiral admolecule. Red arrows placed on the molecular arms indicate the positions of interaction centers responsible for the formation of linear (→←), reversible intermolecular bonds. Equivalent directions of the underlying triangular lattice are marked with intersecting black arrows.

**Figure 2 molecules-30-02656-f002:**
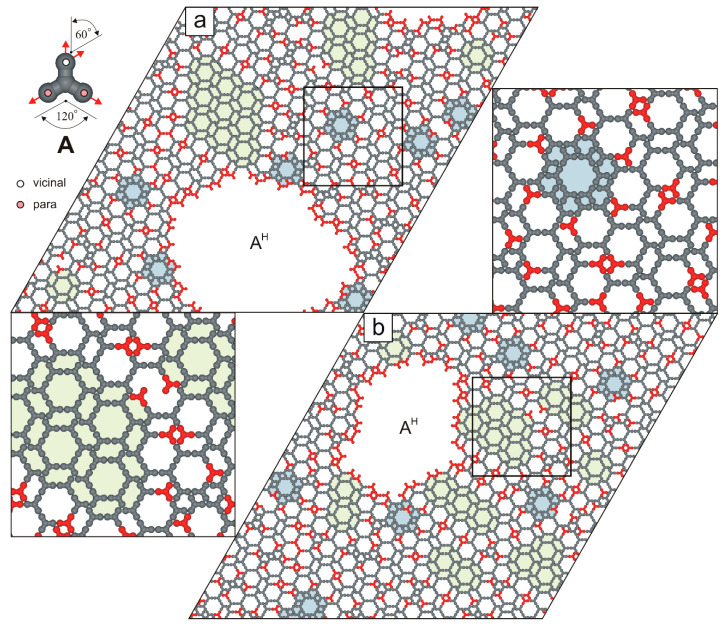
Hybrid overlayer A^H^ composed of (**a**) 1000 molecules **A** (θ=1/3) and (**b**) 1100 molecules **A** (θ=0.36). Under-coordinated molecules **A** are highlighted in red. Isolated floral motifs embedded within the aperiodic phase A^H^ are shaded in pale blue, while small ordered domains are marked in pale green. The top-left inset shows molecule **A** with a color-coded distribution of terminal interaction centers.

**Figure 3 molecules-30-02656-f003:**
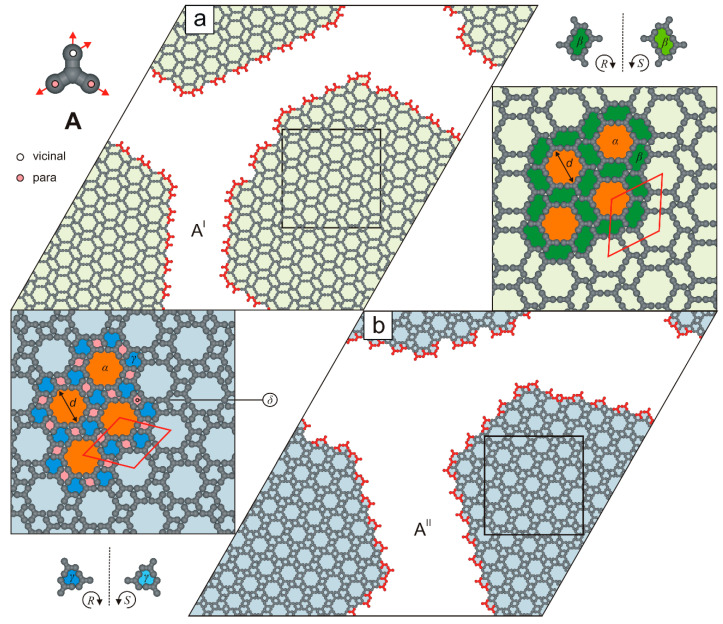
(**a**,**b**) Ordered polymorphs A^I^ (shaded pale green) and A^II^ (shaded pale blue), each comprising 800 molecules **A** (θ=0.26). Rhombic unit cells are outlined with solid red lines. Under-coordinated molecules **A** are highlighted in red. The top-left inset shows molecule **A** with a color-coded distribution of interaction centers. Dashed black lines indicate planes of symmetry. Pores α, β, γ, and δ are differentially color-coded.

**Figure 4 molecules-30-02656-f004:**
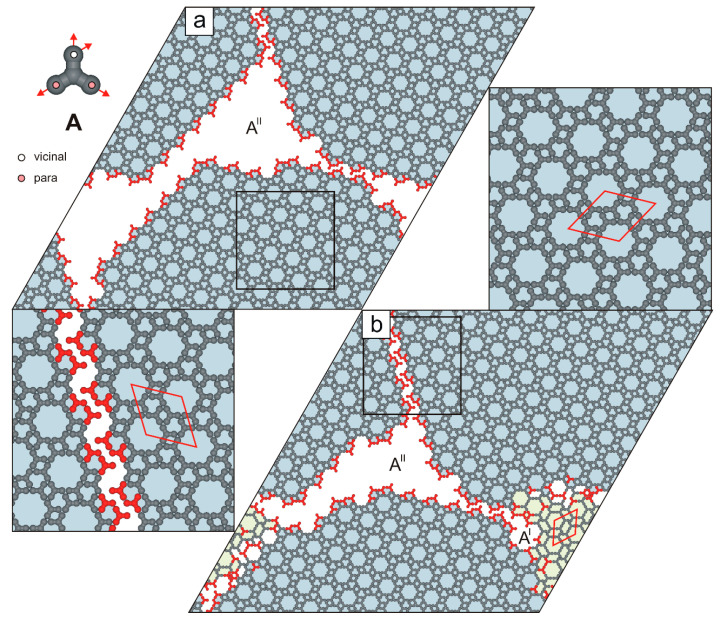
(**a**) Extended network A^II^ comprising 1200 molecules **A** (θ=0.40). (**b**) Coexistence of the network A^II^ with a residual phase A^I^ in a system comprising 1300 molecules **A** (θ=0.43). Under-coordinated molecules **A** are highlighted in red. The top-left inset shows molecule **A** with a color-coded distribution of interaction centers. Rhombic unit cells are outlined with solid red lines.

**Figure 5 molecules-30-02656-f005:**
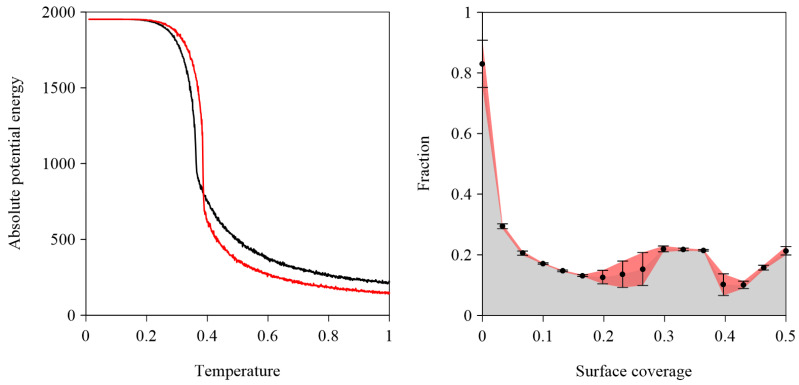
(**left panel**) Effect of temperature on the absolute potential energy of the competing polymorphs A^I^ (solid red line) and A^II^ (solid black line) illustrated in Figure 3. The presented data are averages from 10 independent system replicas. (**right panel**) Effect of surface coverage θ on the fraction f of under-coordinated molecules **A**. The black dots represent averages from ten independent system replicas for each θ value. Vertical black bars indicate standard deviations.

**Figure 6 molecules-30-02656-f006:**
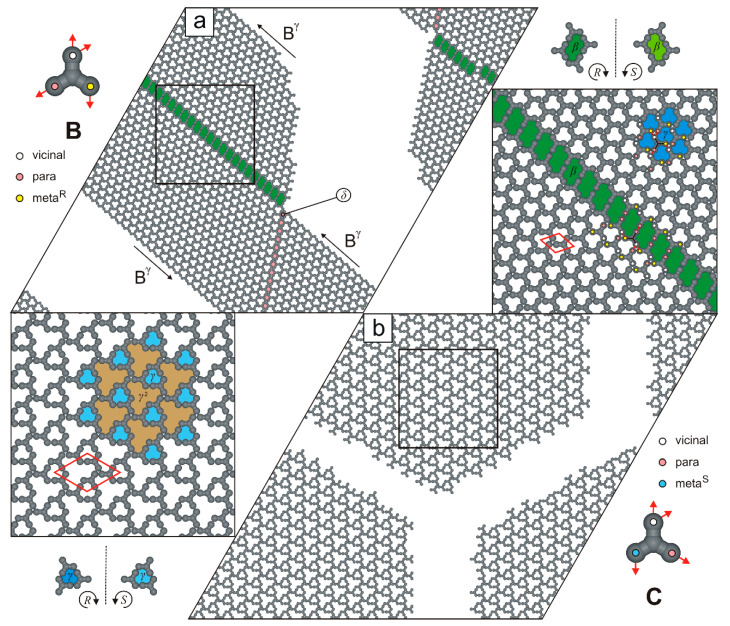
Adsorbed overlayers comprising (**a**) 1000 molecules **B**, and (**b**) 1000 molecules **C**. Molecules **B** and **C** with a color-coded distribution of interaction centers are shown in the top-left and bottom-right insets, respectively. Solid red lines outline the rhombic unit cells of the porous networks, while dashed black lines denote symmetry planes. The differently sized and shaped pores are marked by different colors.

**Figure 7 molecules-30-02656-f007:**
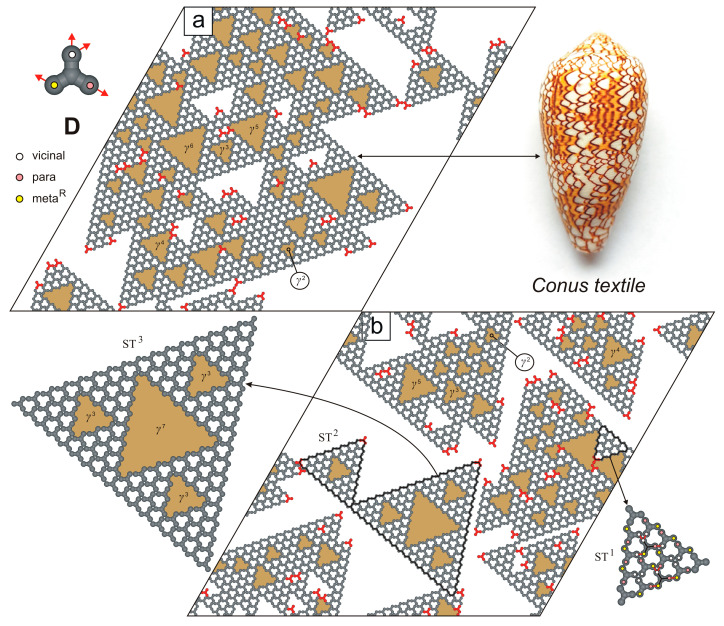
(**a**,**b**) Representative snapshots of adsorbed overlayers comprising 1000 molecules **D**, each. Under-coordinated molecules **D** are highlighted in red. The *Conus textile* sea snail shell is shown in the top-right inset. Randomly distributed nanopores $\gamma^{n}$ (where n=2–7) are marked in gold. The dashed black line indicates a plane of symmetry. The top-left inset shows molecule **D** with a color-coded distribution of interaction centers.

**Figure 8 molecules-30-02656-f008:**
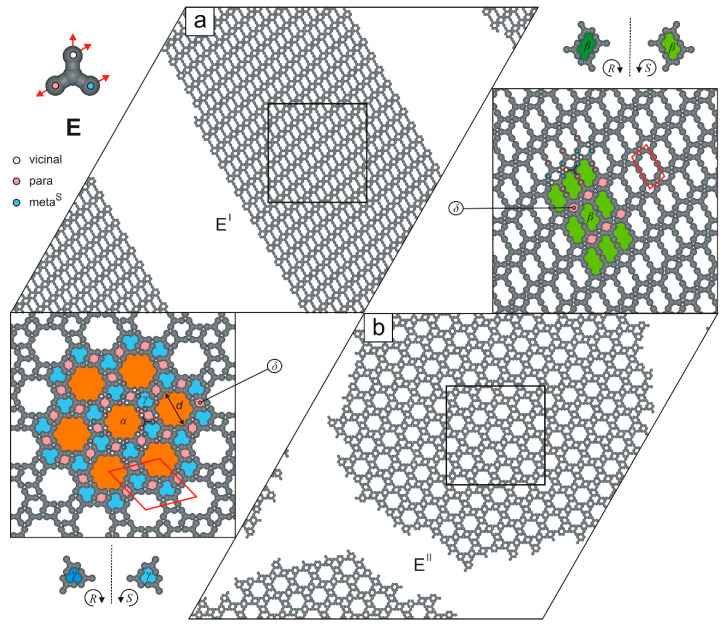
Ordered polymorphs (**a**) E^I^ and (**b**) E^II^ comprising 1000 molecules **E**, each. The top-left inset shows molecule **E** with a color-coded distribution of interaction centers. Solid red lines outline the networks’ unit cells, while dashed black lines indicate planes of symmetry. Pores α, β, γ, and δ are color-coded.

**Figure 9 molecules-30-02656-f009:**
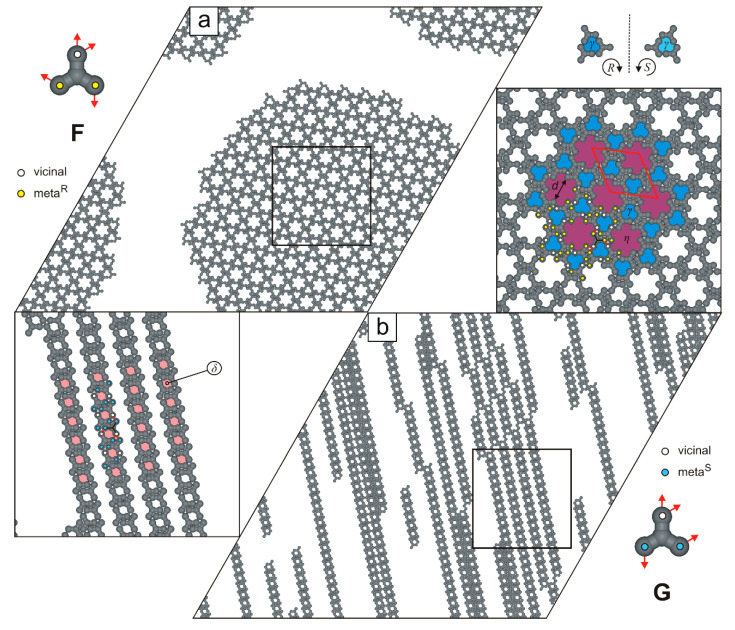
Adsorbed overlayers comprising (**a**) 1000 molecules **F** and (**b**) 1000 molecules **G**, respectively. Molecules **F** and **G**, with color-coded distribution of interaction centers, are shown in the top-left and bottom-right insets, respectively. A solid red line outlines the rhombic unit cell, while the dashed black line indicates a plane of symmetry. Pores γ, δ, and η are color-coded.

**Figure 10 molecules-30-02656-f010:**
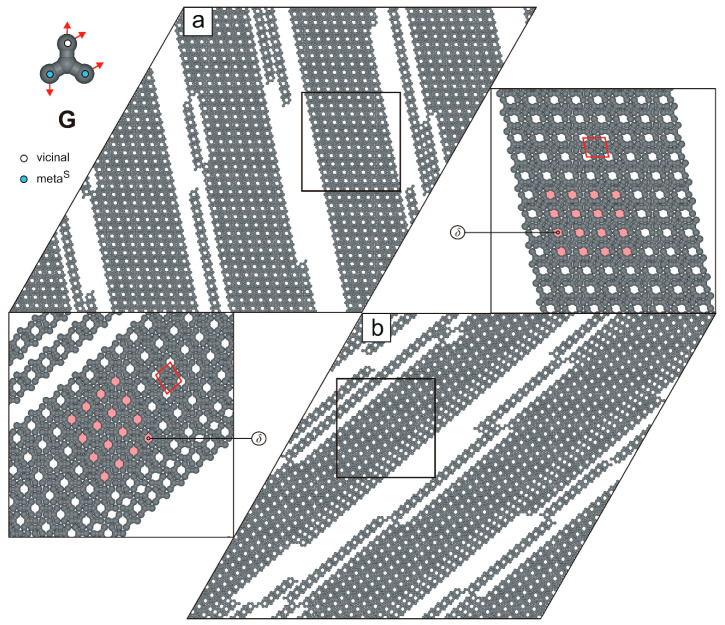
(**a**,**b**) Systems composed of 2000 molecules **G**, each (θ≈2/3). The top-left inset shows molecule **G** with a color-coded distribution of terminal interaction centers. Solid red lines outline unit cells, and the periodically distributed pores δ are highlighted in pink.

**Figure 11 molecules-30-02656-f011:**
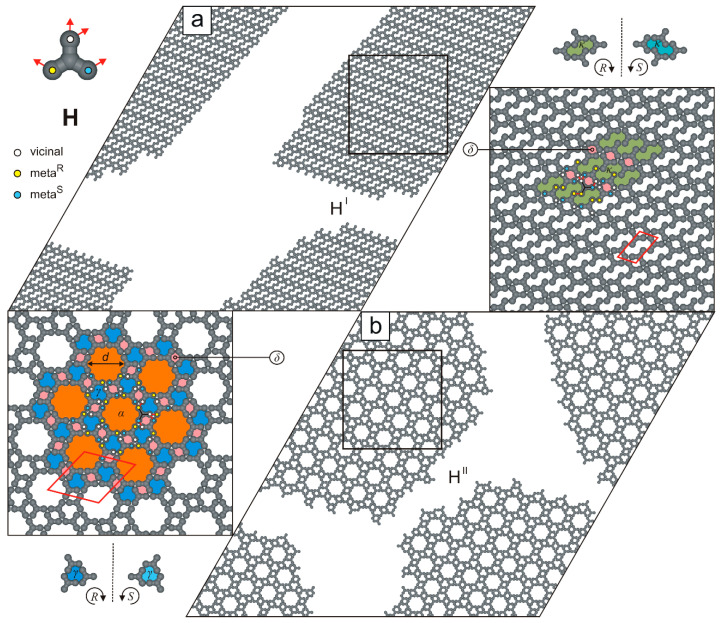
(**a**,**b**) Ordered polymorphs comprising 1000 molecules **H**, each. The top-left inset shows molecule **H** with a color-coded distribution of interaction centers. Solid red lines outline unit cells of the networks, and dashed black lines indicate planes of symmetry. Pores γ, δ, and κ are color-coded.

**Figure 12 molecules-30-02656-f012:**
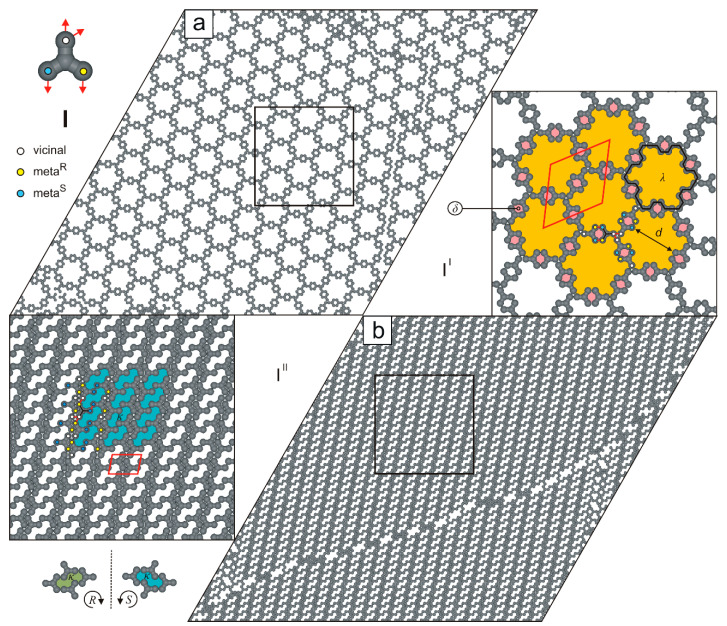
(**a**) Openwork network I^I^ comprising 1000 molecules **I**. (**b**) Brickwall-like overlayer composed of 2000 molecules **I** (θ≈2/3). The top-left inset shows molecule **I** with a color-coded distribution of interaction centers. Solid red lines outline unit cells; the solid black line highlights a large cogwheel-like void λ. The dashed black line indicates a plane of symmetry. Pores δ, κ, and λ are color-coded.

**Figure 13 molecules-30-02656-f013:**
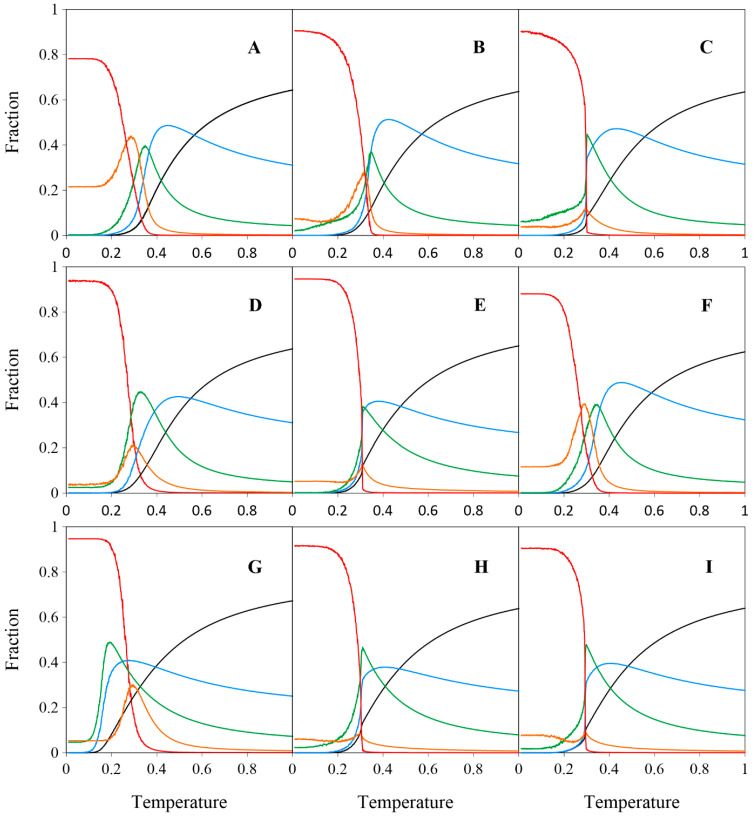
Effect of temperature T on the fraction of molecules **A**–**I** with n=0–4 neighbors in the studied systems (θ≈1/3). For each molecule, data are averaged over 10 independent system replicas. Color codes: n=0 (black), n=1 (blue), n=2 (green), n=3 (orange), and n=4 (red).

**Figure 14 molecules-30-02656-f014:**
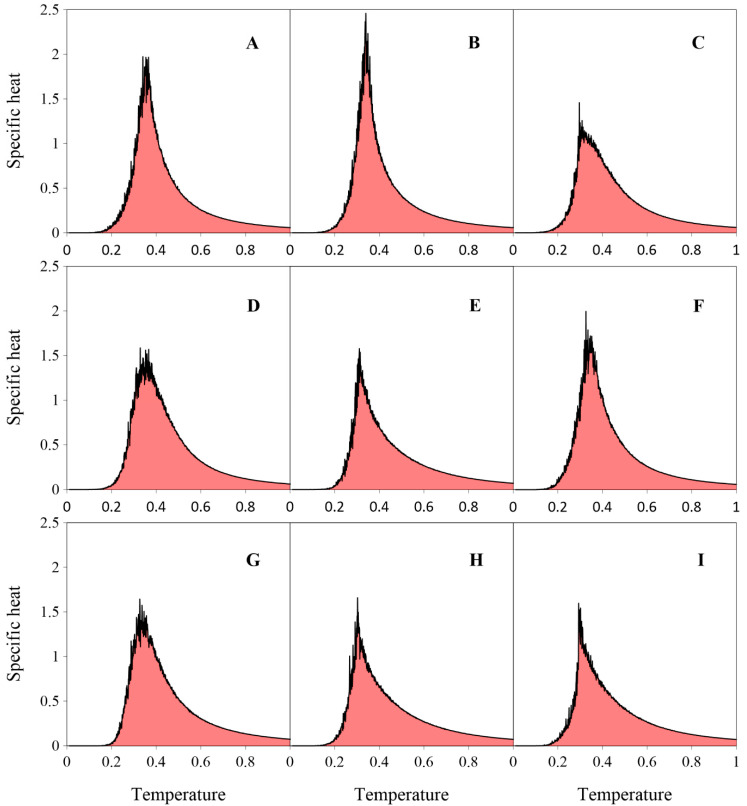
Specific heat c calculated for systems comprising 1000 molecules (**A**–**I**) (θ≈1/3). For each molecule, data are averaged over 10 independent system replicas.

**Figure 15 molecules-30-02656-f015:**
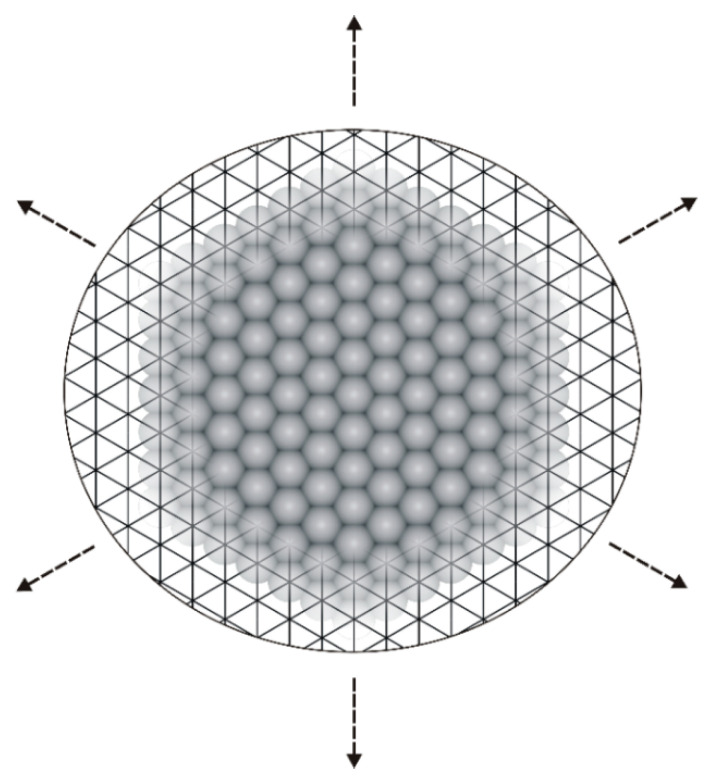
Triangular lattice representing a smooth metallic surface: Cu(111), Ag(111), or Au(111). Dotted arrows indicate the periodic boundary conditions imposed on the lattice.

**Table 1 molecules-30-02656-t001:** Tabulated absolute total potential energies of the hybrid phase A^H^ and polymorphic networks A^I^ and A^II^, each comprising 1000 molecules **A**. The values are averaged over ten independent replicas of the simulated systems. To simulate the self-assembly of the polymorphic networks A^I^ and A^II^ at θ=1/3, the formation of *para–para*/*vicinal–vicinal* intermolecular bonds was blocked.

Replica No.	$\left|\langle E^{H}\rangle \right|$	$\left|\langle E^{I}\rangle \right|$	$\left|\langle E^{II}\rangle \right|$
1	1888	1952	1952
2	1891	1952	1953
3	1893	1953	1952
4	1892	1953	1952
5	1890	1952	1953
6	1890	1952	1952
7	1887	1953	1952
8	1888	1953	1952
9	1889	1952	1952
10	1892	1952	1952
Average <>	1890.00	1952.40	1952.20
Standard deviation	1.90	0.49	0.40
Range	6.00	1.00	1.00

**Table 2 molecules-30-02656-t002:** Statistics of under-coordinated molecules in ten independent replicas of the adsorbed system comprising 1000 molecules **D**.

Replica No.	Under-Coordinated Molecules D	%
1	59	5.90
2	64	6.40
3	66	6.60
4	59	5.90
5	58	5.80
6	60	6.00
7	67	6.70
8	64	6.40
9	61	6.10
10	54	5.40
Average	61.20	6.12
Standard deviation	3.82	-
Range	13	-

## Data Availability

The data contained in this work are available for review by contacting the authors.

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
