# Peer review of "Surface-Confined Self-Assembly of Star-Shaped Tetratopic Molecules with Vicinal Interaction Centers"

_molecules, 2025, doi:10.3390/molecules30122656_

Round 1
Reviewer 1 Report
Comments and Suggestions for Authors
Authors have conducted a computational study on the 2D self-assembly of star-shaped molecular tectons with vicinal functional groups. In total, nine positional isomers (A-I) are considered, each of them equipped with two vicinal interaction centers and two interaction centers at para or meta positions. This coarse-grained model of tectons with anisotropic reversible intermolecular interactions on a triangular lattice corresponds to assemblies of real organic molecules with nitrogen-containing functional groups, adsorbed on fcc metallic surfaces.
Monte Carlo simulations using the model were performed to predict possible supramolecular phases for all tectons, and to study their structural and thermodynamic properties. Authors have obtained in simulations a large variety of 2D structures, namely, homoporous and heteroporous networks, linear molecular strands, Sierpinski-fractal-like aggregates, and aperiodic mosaics. Coordination, density, and porosity was investigated for each structure, and detailed coordination saturation statistics was calculated for the selected cases.
Overall, the article is well written, and the research statements are supported by the results.
I have the following minor comments and questions.
(1) Page 7, line 138: "The discussed overlayer comprises three adjacent, uniporous networks Bγ ...". It seems that all three networks/domains (Figure 6a) can be assigned to the same type of structure?
(2) The same page, line 139: "... which as differently oriented and interconnected by the rows of R-chiral voids beta and achiral cavities delta." The latter phrase is somewhat unclear ("which as" -> "which were"?).
(3) Are polymorphs EI and EII (Figure 8) of the molecular tecton E obtained under the same simulation conditions? I.e. are they competing phases with similar chances of occurrence?
(4) Figure 11a caption mentions rhombic unit cells, but it looks like they are not strictly rhombic due to unequal sides.
(5) Page 17 Line 293: "To eliminated..." -> "To eliminate".
(6) Page 17 Line 297 (The Model and Calculations): the value of ΔT is given as 0.01. Is this correct?
Reviewer 2 Report
Comments and Suggestions for Authors
This manuscript presents Monte Carlo simulations of self-assembled networks formed by various star-shaped molecules with vicinal interaction centers. The study observes different architectures, including two-dimensional crystals, aperiodic mosaics, Sierpiński-like aggregates, and one-dimensional strands. I recommend it published in Molecules after considering the following issues.
1. Are there any experimental results of molecules with vicinal interaction centers? If so, the authors should cite the corresponding references.
2. Are the flips of the molecules, which change their chirality, taken into account during the simulation?
3. AII appeared 72% of all simulated systems. Is it possible due to the energy difference, following the Boltzmann distribution, where the probability is proportional to exp(-E/kT)?
4. In the Figures of the MC results, such as Figure 3, the authors present “R” and “S” pores. However, the enlarged simulated networks (the top-right and the lower-left insets) only show “R” pores. How about the “S” pores? The authors could include another type of pore in the supporting information.
Additionally, the lower-left inset highlights some of “R” pores, while other “R” pores are marked in pale blue, which is the same color as the “S” pores. It may lead to misunderstanding. Please change the color or clearly mark all the pores.
Furthermore, in the top-right inset, the green in “S” pores and the green in AI network looks similar.
5. The discussion of Figure 14 is insufficient, covering only lines 268 to 269. Moreover, the equation to calculate specific heat differs from the previous references, such as, J. Phys. Chem. C 2020, 124, 20066−20078. Why is it divided by 4?
6. During the simulation, the temperature decreases from 1.01 to 0.01. How do the authors make sure that the system reaches a thermodynamically stable state at a specific temperature to obtain the fraction in Figure 13 or the energy for calculating specific heat in Figure 14?
Minor points:
1. The positions of the label (i.e., a and b) in Figures 2 to 4 and 6 to 12 looks unconventional. Typically, they should be located at the corners of each figure.
2. Line 277, 305 and Line 314, there should be no indentation before “where”.
3. Line 309 to 310, there should be no line break.
4. Line 319, there should be a space between “approximately” and “16 hours”.
Reviewer 3 Report
Comments and Suggestions for Authors
This paper uses Monte Carlo computer simulations for predicting the topology of adsorbed overlayers formed by star-shaped tetratopic molecules with vicinal interaction centers. Overall, the paper was organized with reasonable clarity. I think minor revision is needed before publication. The detailed comments are as follows:
- It is recommended to introduce the development of Monte Carlo computer simulations for predicting the vicinal molecular interactions.
- The Figure numbers are noted wrong in lines 222 and 226. It should be Figure 11.
- It may be better to give one or two real examples to prove the simulated results.
Reviewer 4 Report
Comments and Suggestions for Authors
This article proposes plausible topological patterns made of star-shaped tetratopic molecules with vicinal interaction centers, based on lattice Monte Carlo computer simulations. Various types of architectures including two-dimensional crystals, aperiodic mosaics, Sierpiński-like aggregates, and one-dimensional strands are represented on the lattice surface. Although the research content is not interesting, I do not see any particular problems with the description of the paper. I consider this paper acceptable for publication in Molecules.
